# Detecting Orthostatic Intolerance in Long COVID in a Clinic Setting

**DOI:** 10.3390/ijerph20105804

**Published:** 2023-05-12

**Authors:** Robert Oliver Isaac, Joanna Corrado, Manoj Sivan

**Affiliations:** 1National Demonstration Centre for Rehabilitation, Leeds Teaching Hospitals NHS Trust, Leeds LS7 4SA, UK; 2Long COVID Rehabilitation Service, Leeds Community Healthcare Trust, Leeds LS6 1PF, UK; 3Academic Department of Rehabilitation Medicine, Leeds Institute of Rheumatic and Musculoskeletal Medicine, University of Leeds, Leeds LS7 4SA, UK

**Keywords:** post-COVID-19 syndrome (PCS), post-COVID-19 condition (PCC), post acute COVID-19 syndrome (PACS), PoTS, postural hypotension, dysautonomia, C19-Yorkshire rehabilitation scale (YRS)

## Abstract

Introduction: A likely mechanism of Long COVID (LC) is dysautonomia, manifesting as orthostatic intolerance (OI). In our LC service, all patients underwent a National Aeronautics and Space Administration (NASA) Lean Test (NLT), which can detect OI syndromes of Postural Tachycardia Syndrome (PoTS) or Orthostatic Hypotension (OH) in a clinic setting. Patients also completed the COVID-19 Yorkshire Rehabilitation Scale (C19-YRS), a validated LC outcome measure. Our objectives in this retrospective study were (1) to report on the findings of the NLT; and (2) to compare findings from the NLT with LC symptoms reported on the C19-YRS. Methods: NLT data, including maximum heart rate increase, blood pressure decrease, number of minutes completed and symptoms experienced during the NLT were extracted retrospectively, together with palpitation and dizziness scores from the C19-YRS. Mann-Witney U tests were used to examine for statistical difference in palpitation or dizziness scores between patients with normal NLT and those with abnormal NLT. Spearman’s rank was used to examine the correlation between the degree of postural HR and BP change with C19-YRS symptom severity score. Results: Of the 100 patients with LC recruited, 38 experienced symptoms of OI during the NLT; 13 met the haemodynamic screening criteria for PoTS and 9 for OH. On the C19-YRS, 81 reported dizziness as at least a mild problem, and 68 for palpitations being at least a mild problem. There was no significant statistical difference between reported dizziness or palpitation scores in those with normal NLT and those with abnormal NLT. The correlation between symptom severity score and NLT findings was <0.16 (poor). Conclusions: We have found evidence of OI, both symptomatically and haemodynamically in patients with LC. The severity of palpitations and dizziness reported on the C19-YRS does not appear to correlate with NLT findings. We would recommend using the NLT in all LC patients in a clinic setting, regardless of presenting LC symptoms, due to this inconsistency.

## 1. Introduction

Long COVID (LC) is a patient-derived term for persistent symptoms >4 weeks after COVID-19 infection [1]. LC includes both ongoing symptomatic COVID-19 (>4 weeks since infection) and post- COVID-19 syndrome (>12 weeks since infection) [1]. LC involves clusters of multisystemic symptoms that may fluctuate or change over time, and common symptoms include fatigue, breathlessness, pain, brain fog, palpitations and dizziness [1]. The long-term effects of SARS-CoV2 infection are estimated to affect the day-to-day activities of 1.6 million people in the UK [2].

There is an increasing body of evidence suggesting a high prevalence of dysautonomia and orthostatic intolerance (OI) in LC [3,4,5,6]. Postural Tachycardia Syndrome (PoTS) and Orthostatic Hypotension (OH) are OI syndromes that can be detected in LC [1]. Outside of the LC setting, a negative impact on work ability and functioning was found in patients with PoTS compared with healthy controls [7].

One of the mechanisms of LC has been described as an immune-mediated dysfunction of the autonomic nervous system, which may result in OI [8]. This could be brought about through pro-inflammatory cytokine release during acute COVID-19 infection, or due to an autoimmune process [8]. Reduced activity may lead to further exacerbation of OI, as prolonged bed rest has shown to impair baroreflex adjustments in healthy volunteers [9]. Autonomic dysfunction in LC remains idiopathic and should be distinguished from known peripheral or central structural conditions leading to autonomic dysfunction [3,4,5,6,10].

Guidance from the National Institute of Clinical Excellence for the management of patients with LC recommends investigating for OI in symptomatic patients, using a leaning or standing test. These tests can identify patients likely to have Postural Tachycardia Syndrome (PoTS) or Orthostatic Hypotension (OH) [1]. The Leeds LC rehabilitation service uses the National Aeronautics and Space Agency (NASA) Lean Test (NLT), which has been used in conditions such as Myalgic Encephalomyelitis/Chronic Fatigue Syndrome (ME/CFS) and fibromyalgia [11]. It involves a series of blood pressure (BP) and heart rate (HR) measurements, initially for 2 min of lying, followed by 10 min of standing (leaning against the shoulder blades with the heels six inches from the wall) [12]. Another similar screening test used by centres is the active stand test [5]. A further home test recently reported in the literature is the aAP (adapted Autonomic Profile), which is a series of short lean tests measuring fluctuations in BP and HR in the context of activities which may precipitate LC symptoms, such as physical activity, food and mental exertion [10].

These simple tests can be completed in a clinic setting, at the bedside or even at the patient’s home environment, and can be initiated with a relatively modest amount of training [10,11]. Given the prevalence of LC, these can provide a highly practical method to detect OI. In contrast, other more specialised tests for diagnosing OI in the context of LC include head up tilt (HUT) testing and cerebral blood flow (CBF) through transcranial Doppler [6]. As it provides a more direct measurement of cerebral perfusion, which is assumed to lead to symptoms of OI, CBF may be a more sensitive test for OI [13,14]. Significant postural changes in CBF have been found in patients with LC or ME/CFS on HUT even in the absence of significant HR or BP changes [6,14]. However, access to HUT or CBF is limited to large hospital and research settings and is not feasible to be used in all patients in a clinical setting and in Low- and Middle-Income Countries (LMIC).

Symptoms of LC can be captured on Patients Reported Outcome Measures (PROMs). As LC is a novel multisystem condition, the recommendation is to use condition-specific PROMs [15]. There is still a lack of consensus on the ideal outcome measure for symptoms of LC. Two new measures that have been developed and validated in the condition include the COVID-19 Yorkshire Rehabilitation Scale (C19-YRS) [16] and the Symptom Burden Questionnaire (SBQ-LC) [17].

The objectives of this retrospective study were (1) to report on the findings of the NLT in patients with LC under a specialist community service and (2) to examine the correlation between NLT findings and patient-reported LC symptoms on the C19-YRS.

## 2. Materials and Methods

This study was carried out in the Leeds LC rehabilitation service specialist clinics. Patients in this service had severe persistent LC symptoms impacting on daily functioning for at least 3 months and were referred to the service by primary care physicians. To be treated under the service, patients require a confirmed COVID-19 diagnosis, or illness consistent with COVID-19 for patients who became unwell prior to availability of mass testing. Patients required basic investigations for symptoms to be organized by the primary care physician. This was mainly for the purposes of diagnosing conditions other than LC which might be contributing to symptoms. One example would be investigating for breathlessness with a chest X-ray and an electrocardiogram.

Unless contraindicated, all patients under the service underwent an NLT for screening of OI, which was carried out by 2 technical assistants. BP and HR measurements are taken for 2 min in lying position. Following this, patients are asked to stand, leaning with their shoulder blades against the wall with the heels six inches from the wall, and measurements of BP and HR were taken every minute for up to 10 min [8]. Any symptoms experienced during the NLT, such as OI symptoms, were recorded. Patients who experienced red flag symptoms, or who had NLT abnormalities, discussed these with the medical team, which included medical doctors in cardiology, respiratory and rehabilitation medicine. Patients also completed a C19-YRS, which includes scores for their current symptoms, and symptoms they were experiencing before their COVID-19 infection.

In our study, a sample of 100 consecutive patients who had undergone an NLT were selected retrospectively. All patients had consented for anonymous use of their data for research, and the clinical database “system one” electronic health records were used to obtain further demographic data. To be eligible for analysis, patients required a fully completed initial C19-YRS, which includes palpitation and dizziness severity scores, as well as at least one set of blood pressure (BP) and heart rate (HR) measurements in both lying and standing positions for the NLT.

Data for the maximum increase in HR and decrease in systolic blood pressure (SBP), diastolic blood pressure (DPB), number of minutes of NLT completed and symptoms experienced during the NLT were recorded. Haemodynamic criteria for PoTS and OH as defined in the literature were used for the analysis of findings (Table 1) [13,18]. We additionally analysed upper limits of normal values of HR by subcategorizing patients with a lying to leaning increase of 25 BPM (Tachycardia 25, T25) and 20 BPM (T20).

The C19-YRS palpitations and dizziness scores (0–10 Likert scale) were modified into 4 response categories (no/mild/moderate/severe) as described in the C19-YRS literature which allowed for better correlation analysis [19]. Statistical analysis of data was carried out using SPSS IBM. Mann-Witney U tests were used to examine whether patients with abnormal NLT findings (PoTS or OH) had significantly different OI scores (palpitation and dizziness) on the C19-YRS, compared to those with normal NLT. We additionally used Spearman’s rank to examine for any correlation between postural HR or BP change and OI scores. We additionally compared the results by patient gender to examine for any association with postural HR or BP change, or symptom severity scores.

## 3. Results

### 3.1. Demographics

Data sets for 100 patients at the Leeds LC service who had an NLT between August 2021 and March 2022 were analysed. Table 2 shows the detailed demographics of the group analysed, where a majority of the participants were women, and only 16 out of 100 participants required hospitalisation for acute COVID-19, suggesting mainly a mild acute illness in the majority. The mean length between the onset of COVID-19 diagnosis and NLT was 427 days (range 144–945 days).

### 3.2. NASA Lean Test (NLT) Abnormalities

#### 3.2.1. Heart Rate and Blood Pressure

Table 3 shows the mean change in HR from lying to leaning during NLT. Of note, there is a large variation in the standard deviation and range for all values. No significant differences in mean HR, DBP or SPB was found when comparing patients taking antidepressants, antihypertensives or hormonal medication.

Table 4 shows the NLT data when split by gender. There was no statistical difference between men and women in any of the NLT results.

In total 13 of the 100 patients (10 female and 3 male) showed haemodynamic trends consistent with PoTS at the sustained 30 BPM threshold (T30). In addition, 10 (8 female and 2 male) had an upper limit of normal value tachycardia at T25, and an additional 18 (12 female and 6 male) at T20. A total of 41 patients therefore showed a sustained HR increase by at least 20 BPM. Figure 1 shows the frequency distribution of the maximum sustained HR. Nineteen patients had a (non-sustained) maximum rise of 30 BPM.

Nine patients (eight female and one male) met the haemodynamic criteria for OH during the NLT. In five patients, the BP decrease was in both SBP and DBP, three for DBP only and one for SBP. Furthermore, 30 patients (18 female, 12 male) had delayed OH. Of these, 17 were for SBP, 7 for both SBP and DBP, and 6 for DBP.

#### 3.2.2. Non-Tolerance of NLT and Symptoms Experienced

Overall, 25 out of the 100 patients (15 female, 10 male) were not able to complete a full 10 min of NLT due to overwhelming symptoms (distribution in Figure 2). A total of 15 out of the 25 had symptoms suggestive of OI, such as dizziness, light headedness, faintness or feeling hot or clammy. Additionally in the group not tolerating a full NLT, three met the criteria for OH, six for delayed OH, one for PoTS at the T30 threshold, two at T25 and three at T20. There were symptoms of OI recorded during NLT for 3 out of the 9 patients who met the criteria for OI, 5 of 13 for PoTS, 10 out of 23 for T25 and 13 out of 41 at T20.

### 3.3. Symptom Severity as Reported on the C19-YRS and Comparison with NLT

There was no significant difference between mean total C19-YRS, dizziness or palpitation scores by gender (Table 5). On the C19-YRS, 81 out of the 100 participants reported dizziness, with 68 out of 100 for palpitations. Table 6 and Table 7 show the severity of dizziness and palpitation scores, broken down by NLT haemodynamic subgroups.

There was no correlation between the raw dizziness or palpitation severity scores and degree of haemodynamic abnormality (heart rate increase or blood pressure decrease) in the NLT (Table 8).

Table 9 shows the results of Mann-Witney U testing, with no significant changes between the medians of palpitations and dizziness scores when broken down by NLT category.

## 4. Discussion

In this population of patients with ongoing LC symptoms, we have found evidence of OI, both symptomatically and haemodynamically detected on the NLT. In total, 38% of patients reported symptoms of OI on the NLT, and symptoms of OI were more frequent in those patients unable to tolerate the NLT. On the NLT, 13 patients met conventional haemodynamic criteria for PoTS, a further 28 patients had postural tachycardia on the upper limits of normal at T25 and T20 and 9 patients met the criteria for OH. The C19-YRS data suggest that there is lack of correlation between symptom severity and findings from the NLT, suggesting lack of predictive ability for targeting NLT towards individuals with specific symptoms. We have found no significant association between gender and NLT findings or C19-YRS results.

The prevalence of OI in the general population is believed to vary considerably by age and gender, estimated to be between 6 and 35% for OH [20] and about 0.2% for PoTS [13]. Given the high rates of symptoms and physiological abnormalities in this LC group, we would suggest that an OI test is performed all LC patients. The high non-completion rate for the NLT is relevant, as a shorter test may lead to an underdiagnosis of PoTS [11]. Of note we found a significantly higher report of postural symptoms in the group who were unable to complete a full 10 min of NLT.

The high number of abnormal results in our study is comparable to other studies which suggest that OI is relatively common in patients suffering from LC [3,4,5]. In a cross-sectional study of 85 patients undergoing a 3 min stand test followed by a 10 min head up tilt table test, Monaghan et al. found that 66% of participants had symptoms of OI on active stand, but failed to demonstrate OH or PoTS as a predictor for this [21]. In a prospective longitudinal study of 24 patients with LC with known OI symptoms, Jamal et al. found that nearly all patients had evidence of autonomic dysfunction on head up tilt table testing [4]. In a prospective study of 180 patients, Stella et al. found significant questionnaire symptom-based evidence of dysautonomia, and found that 13.8% of participants had OH, but none had PoTS, though they only took measurements for 3 min of standing [5]. Vernon et al. demonstrated a significant worsening of fatigue and brain fog symptoms during the time of NLT in patients with LC and ME/CFS, which mainly improved after 2 days [3]. In this study, 23% of LC patients had PoTS, and 2% had OH, and there were significant haemodynamic differences between healthy controls and patients with LC [3].

Alternative screening for OI could include the adapted Autonomic Profile (aAP), which can be used by patients to capture heart rate and blood pressure data themselves in a home setting [10]. Symptoms of LC are known to fluctuate and may not fully be captured in a one-off bedside physiological test carried out over 15 min. In contrast, the aAP screening test, which enables multiple “mini” NLTs to be repeated over different points in a patient’s day, may provide more data and can be carried out by patients in their own setting, potentially when symptoms are worse [10]. There are many advantages of this profile, including being easily repeatable and helping to establish links of symptoms with possible triggers, which may be more informative for patients. In our study, the severity of dizziness and palpitation symptoms do not appear to match with the degree of haemodynamic change seen in the NLT. The lack of correlation makes a case for using the test in every LC patient.

Transcranial Doppler ultrasound has been used to measure CBF in other settings. In one study of 510 patients diagnosed with ME/CFS, Van Campen et al. demonstrated that OI, as manifested by a significant decrease in cerebral blood flow on postural challenge, was present even in those with no significant HR or BP abnormalities [14]. The same team also described significant decreases in cerebral blood in patients with LC with no diagnosable postural signs [6]. These data show the advantage of more direct measurements of physiological parameters in symptomatic patients. Some of the disadvantages of this test include accessibility on a large scale, and obtaining sufficient image quality may limit the use in certain patients [14].

Our study has a few limitations. Our recording of symptoms from the C19-YRS took place within a month for all patients but would not necessarily represent the symptoms experienced by patients at the time of the NLT. While the C19-YRS is validated as an outcome measurement tool for LC symptoms, it is not a detailed assessment of patients’ postural symptoms. It is worth considering that many other factors can affect the severity of palpitations and dizziness, which may not be captured by the NLT. On the other hand, our study also highlights the fact that it is not advisable to target NLT for only those with specific symptoms, as dysautonomia symptoms are quite variable across the organ systems and can present with a wide array of symptoms, similar to LC [10].

There was no documentation of symptoms recorded during NLT for 50 of the participants. Although this was likely due to them not experiencing symptoms, this may have resulted in underestimating the true symptoms experienced in NLT, with effects on any correlation between haemodynamic values and true NLT symptoms. There are other limitations in our study attributable to our population selection. This sample of patients had severe LC symptoms impacting on daily functioning and had a mean of 427 days between COVID-19 onset and NLT and this ranged from 144 to 945 days. We would expect this group to represent patients with LC symptoms on the more severe end, which may have resulted in abnormalities such as OI being more likely. Due to the relatively small sample size of 100 patients, we were unable to account for all factors that might affect OI. Age has been shown to be a significant factor in the prevalence of OI in the non-LC population [13,20] and we have not had sufficient data to categorise the results by age. Further factors that are likely to impact the results include use of medications such as antihypertensives and beta blockers, which were continued, and anxiety, which was the most common comorbidity in this group. Further evaluation with a bigger sample size is needed to understand the relationship between these symptoms and NLT findings. It would also be useful to compare NLT results with other methods of measuring OI, including HUT and CBF.

## 5. Conclusions

This retrospective study found evidence of OI patients with LC, with impact on daily functioning. The NLT can be used to detect OI in a clinic setting. We would recommend using this test for all patients with LC and not just patients with classical OI symptoms, due to the high prevalence of OI in LC. Further confirmatory tests of OI including HUT and CBF would be ideal; however, accessing these tests in a community setting is difficult. Further work is required to evaluate the sensitivity and specificity of NLT for OI in LC.

## Figures and Tables

**Figure 1 ijerph-20-05804-f001:**
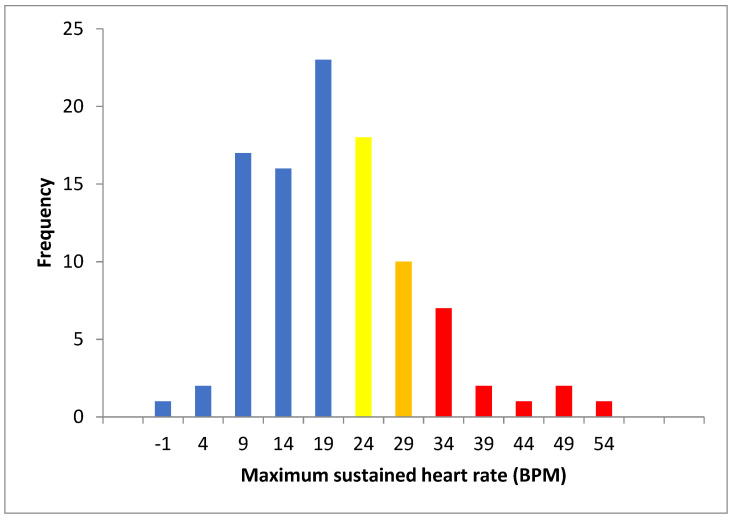
Frequency graph of maximum sustained heart rate change lying to standing (red T30, orange T25 and yellow T20).

**Figure 2 ijerph-20-05804-f002:**
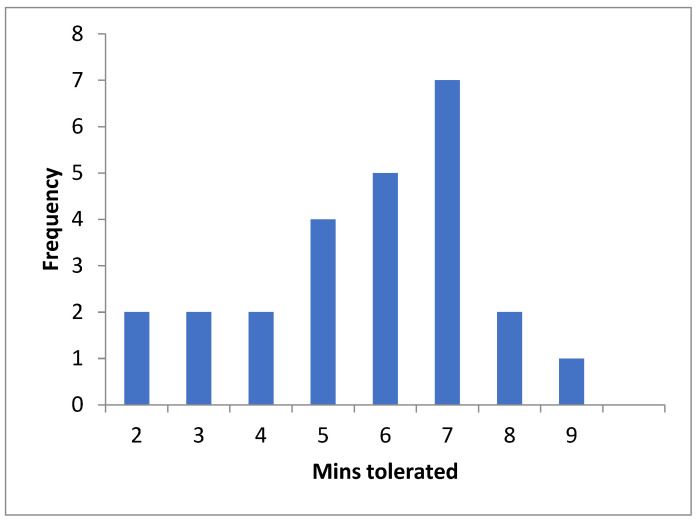
Frequency of number of minutes of NLT tolerated for incomplete NLT.

**Table 1 ijerph-20-05804-t001:** Screening criteria for PoTS and OH used in analysis.

Haemodynamic Postural Tachycardia Syndrome (PoTS) Criteria	
Definition	A increase in heart rate of more than 30 BPM from lying to standing sustained across 2 consecutive readings at least 1 minute apart
Exceptions	An increase of 40 BPM was required in patients up to 18 years of age
Invalid cases	Orthostatic Hypotension
**Haemodynamic Orthostatic Hypotension (OH) Criteria**	
Systolic definition	Decrease in blood pressure of 20 mmHg from lying to standing in the first 3 min
Diastolic definition	Decrease in blood pressure of 10 mmHg from lying to standing in the first 3 min
Delayed OH definition	Systolic or diastolic OH after the first 3 min of standing
**Additional Analysis**	
An increase in heart rate of more than 25 BPM (T25) and 20 BPM (T20) from lying to standing in 2 consecutive readings	

**Table 2 ijerph-20-05804-t002:** Patient demographics.

Gender	
Female	69 (69%)
Male	31 (31%)
**Age**	46.62 (range 17–74)
**Hospitalisation**	
Number hospitalised for acute COVID-19 treatment	16 (16%)
Number admitted to intensive care for acute COVID-19 treatment	4 (4%)
**Ethnicity**	
White	84 (84%)
Asian (includes any Asian background, for example, Bangladeshi, Chinese, Indian, Pakistani)	9 (9%)
Black, African, Black British or Caribbean (includes any Black background)	3 (3%)
Mixed or multiple ethnic groups (includes any Mixed background)	1 (1%)
Unknown	3 (3%)
**Comorbidities**	
Anxiety	27 (27%)
Asthma	18 (18%)
Depression	16 (16%)
Diabetes	8 (8%)
Ex smoker	32 (32%)
Hypothyroidism	3 (3%)
Hypertension	15 (15%)
Ischaemic heart disease	3 (3%)
Migraines	2 (2%)
Perimenopausal	3 (3%)
Smoker (current)	4 (4%)
Other diagnoses not listed above	30 (30%)
**Medications**	
Antidepressant	20 (20%)
Antihypertensive	18 (18%)
Hormonal (contraceptive or hormone replacement therapy)	11 (11%)

**Table 3 ijerph-20-05804-t003:** Mean changes in HR and BP from lying to leaning during NLT.

	Mean	Standard Deviation	Range
Maximum sustained HR change (BPM)	18.45	9.93	−5 to 53
Maximum systolic BP change (mmHg)	−14.11	11.26	−64 to 5
Maximum diastolic BP change (mmHg)	−4.34	8.55	−37 to 13

**Table 4 ijerph-20-05804-t004:** NLT haemodynamic findings by gender.

NLT Heamodynamic Findings	Gender	Mean	Std. Deviation	*p*
Maximum sustained heart rate (BPM)	F	18.88	9.977	0.52
	M	17.48	9.902	
Maximum non-sustained heart rate	F	22.59	10.913	0.65
	M	21.48	11.561	
Maximum systolic BP decrease (mmHg)	F	−14.41	11.446	0.69
	M	−13.45	11.006	
Maximum diastolic blood pressure decrease (mmHg)	F	−4.2	7.079	0.84
	M	−4.65	11.283	
Minutes completed in NLT	F	9.1	1.986	0.23
	M	8.52	2.293	

**Table 5 ijerph-20-05804-t005:** C19-YRS findings by gender.

C19-YRS Results	Gender	Mean	Std. Deviation	*p*
Pre-COVID total C19-YRS total score (out of 100)	F	7.09	8.922	0.74
	M	6.55	6.632	
Initial post COVID C19-YRS total score (out of 100)	F	42.28	16.891	0.58
	M	40.32	16.134	
Dizziness severity score (out of 10)	F	4.22	3.143	0.15
	M	3.32	2.663	
Palpitation severity score (out of 10)	F	3	2.975	0.38
	M	3.58	3.063	

**Table 6 ijerph-20-05804-t006:** C19-YRS dizziness severity scores broken down by haemodynamic diagnoses or PoTS and OH.

Dizziness Severity	Tachycardia				OH (<3 min)	OH (>3 min)	No > T20 Tachycardia or OH
	PoTS at T30	T25	T20	All Tachycardia			
No problem	1	4	4	9	2	6	8
Mild	7	3	7	17	3	14	12
Moderate	5	3	6	14	4	9	1
Severe	0	0	1	1	0	1	5
Total	13	10	18	41	9	30	20

**Table 7 ijerph-20-05804-t007:** C19-YRS palpitation severity scores broken down by haemodynamic diagnoses or PoTS and PH.

Palpitations Severity	Tachycardia				OH (<3 min)	OH (>3 min)	No >20 Tachycardia or OH
	PoTS at T30	T25	T20	All Tachycardia			
No problem	5	7	3	15	3	13	1
Mild	6	2	8	16	3	12	15
Moderate	2	1	4	7	2	5	2
Severe	0	0	3	3	1	0	2
Total	13	10	18	41	9	30	20

**Table 8 ijerph-20-05804-t008:** Spearman’s rank for NLT haemodynamic values and C19-YRS-reported OI symptoms.

	Dizziness Severity Score	Palpitation Severity Score
HR change	0.001 (*p* = 0.995)	−0.168 (*p* = 0.095)
SPB change	0.013 (*p* = 0.897)	0.160 (*p* = 0.111)
DPB change	−0.083 (*p* = 0.409)	0.083 (*p* = 0.411)

**Table 9 ijerph-20-05804-t009:** Independent Mann-Witney U test for dizziness and palpitation medians.

Comparison Category	Symptom Category	U	*p*
PoTS (sustained 30 BPM) vs. not PoTS	Dizziness	440.5	0.197
	Palpitations	620	0.569
OH vs. not OH	Dizziness	434.5	0.801
	Palpitations	349.5	0.462
OH and delayed OH vs. not OH	Dizziness	1154.5	0.803
	Palpitations	1043	0.292
PoTS (sustained 30 BPM) or OH vs. not PoTS or OH	Dizziness	758	0.402
	Palpitations	852.5	0.963
Syncopal during lean test vs. asymptomatic	Dizziness	1088	0.470
	Palpitations	1123	0.632

## Data Availability

Anonymised data can be obtained by contacting the corresponding author.

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
