# Peer review of "Detecting Orthostatic Intolerance in Long COVID in a Clinic Setting"

_ijerph, 2023, doi:10.3390/ijerph20105804_

Round 1

Reviewer 1 Report (Previous Reviewer 1)

With this reply and alterations, the authors did NOT fulfill the recommendations done. For instance POTS 40 bpm or more is 18 and NOT 19. Discussing cerebral blood flow in orthostatic intolerance complaints with a lot of people not having pots and OH is mandatory and not done. So this manuscript does NOT qualify for publication.

Author Response

Thank you for your comment. We have updated the definition of PoTS of 40 BPM for the age of 18 rather than 19. We agree that cerebral blood flow is an important test in patients with severe symptoms and normal postural BP and HR. Unfortunately, not many clinics around the world will have access to specialised tests such as cerebral blood flow. If there are no simple clinic tests to detect orthostatic intolerance, there is a danger of missing the diagnosis. We have mentioned the cerebral blood flow measurement as a test in the introduction, as well as a limitation of our study. Given the prevalence of the condition of around 2% of the population in the UK, we feel it is imperative to carry out simple clinic-based tests for orthostatic intolerance initially. We acknowledge the limits of this test in our discussion.

Reviewer 2 Report (New Reviewer)

The authors show that the prevalence of orthostatic intolerance (OI) is common in patients with a diagnosis of long Covid. However, the assessment of symptoms of OI by C19-YRS questionnaire symptoms of OI does not correlate with objective hemodynamic measurements from the NLT – BP or HR changes.

Neither the information on the high prevalence of OI in Long Covid nor the lack of correlation of OI symptomatoloy with hemodynamic testing is new. Van Campen et al.. 2022, investigated the development of hemodynamic changes in LC-patients referred to the tilt test because of signs of OI. In this paper patients were screened for OI and then investigated.

Van Campen et al. found that the patients with OI complaints had reduced cerebral blood flow, with changes in HR or BP or without hemodynamic changes. The latter means that symptoms of OI are not necessarily caused by changes in HR (tachycardia) nor by changes in BP (hypotension) which is a reasonable physiological assumption but it is the lower cerebral blood flow that causes symptoms and (counterproductive) counterregulatory responses. This study is not discussed although it is an investigation of OI in long Covid. It should be included and discussed.

 It is a reasonable physiological concept that symptoms of OI are not caused directly by a fall in BP or by a rise in HR. During exercise HR can rise much higher and this does not cause symptoms. BP and HR changes are surrogate markers not directly related to symptoms. The fall in cerebral blood flow is the main mechanism causing symptoms as well as the rise in sympathetic activity in an attempt to compensate and the resulting orthostatic stress.

The appropriate parameter to measure would be cerebral blood flow. Van Campen et al. noted that with the duration of long Covid the phenotype of OI had changed. POTs was prevalent earlier, while with the duration of long Covid there was shift away from POTS towards the phenotype with orthostatic hypotension or the phenotype showing no BP or HR changes, which otherwise define OH or POTS. All these patients showed a decrease in cerebral blood flow.

The phenotype with a decrease in cerebral blood flow but no hemodynamic changes (HR or BP) is the most interesting one to consider. The question why this phenotype exists is important for the recommendations. If vasoconstriction as a response to the orthostatic challenge is strong enough there will be no hypotension (but an increase in cerebral vasoconstriction with a decrease in cerebral blood flow). Chronotropic incompetence has been found in ME/CFS and in long Covid. The mechanisms of chronotropic incompetence may limit the rise in HR so that the criteria for POTS are not fulfilled.

Van Campen’s results and yours show that patients with orthostatic symptoms without BP or HR changes, that otherwise define orthostatic hypotension or POTS, are frequent. These patients would not be recognized in the NLT, but by the measurement of cerebral blood flow. Does it then make sense to recommend the NLT for screening knowing that a considerable part of the long Covid patients suffering from OI would not be recognized? Only with the measurement of cerebral blood flow it would make sense from a scientific point of view. From a practical clinical point of view one can question this recommendation as it is obviously symptomatology of OI that counts.

Page 9, line 226 “co morbidity”

Author Response

Many thanks for your helpful comments and insight. We agree that it would be useful to discuss Van Campen et al’s work, and we have included the lack of availability of CBF measurement as one of the limitations of our work. It is a great shame that this is not currently accessible within our service, and we hope that this might be available in the future for severe symptoms despite normal NLT. It is worth noting that the CBF test as a routine test for all patients will not be available in many countries, particularly low and middle-income countries. If we do not use a simple clinic-based test, there is a risk of missing the diagnosis in millions of individuals with long covid worldwide.

Reviewer 3 Report (New Reviewer)

The authors conducted a retrospective study on a cohort of 100 patients with long Covid. The study involved administering an orthostatic tolerance test, specifically the Nasa lean test, and a C19-YRS questionnaire to the patients. The aim was to investigate the potential link between the Nasa lean test results and reported orthostatic intolerance symptoms on the questionnaire. The authors observed a significant number of patients with an abnormal Nasa lean test, which was not predictable based on the questionnaire results. Based on their findings, the authors recommend that orthostatic tolerance should be systematically evaluated in patients with long Covid. The study is noteworthy due to the inclusion of a large patient population, interesting data, and the demonstration that bedside assessment of orthostatic tolerance can be easily performed in this population.

The study, however, could be improved based on major and minor comments.

Major

One major limitation of the study is Berkson's bias, as the population that sought medical evaluation or advice at the long Covid service may not be representative of the overall population of long Covid patients. The authors should acknowledge this point and provide a more precise description of the population. For example, they should indicate whether the 100 patients were consecutive patients or how they were selected, how Covid was confirmed, how they defined severe ongoing long Covid symptoms (line 66), and what the delay was between the Covid diagnosis and the visit to the long-Covid service.

The authors used a simple test to assess orthostatic tolerance, which can be performed in any medical center. This test seems appropriate for evaluating the large number of patients with long Covid and contrasts with the highly specialized tests typically used to assess orthostatic tolerance. The authors should discuss this point.

It is important to clarify for readers that there is a difference between suspected dysautonomia in patients with long Covid, based on symptoms related to the autonomic nervous system, and a demonstrated dysautonomia diagnosed using tools such as the Ewing's score.

The results section includes a description of findings according to gender, but this is not described in the methods section and does not match the study objectives. The authors should address this inconsistency.

Lastly, the term "cohort" is used twice in the manuscript (L 175 & L230), but it may not be appropriate as it refers to a specific biomedical research design. It is recommended to avoid using this term.

Minor

The authors should clearly mention early in the manuscript and in the abstract that the study is retrospective.

In lines 17 and 18, the authors mention statistical analyses, but the reader does not know what was compared. A correlation is not a comparison. Mann-Whitney's test is for unpaired data while the same patient performed NLT and C19-YRS. More generally, the statistics raise some questions: Mann-Whitney's test is for numerical data, not for categories. Pearson's correlation is parametric while Mann-Whitney's test is nonparametric.

Line 20: What does PH mean? (OH?)

Palpitations could be seen in cases of PoTS, but they could also indicate a serious cardiac disease. How were the patients with palpitations found on the C19 YRS questionnaire managed?

Table 3 mentions a maximum sustained HR change of -5 BPM. This finding is surprising in this population and could indicate a mistake or a severe dysautonomia. How were the patients with this finding managed?

Lines 123 and 124: Please specify whether the OH was systolic, diastolic, or both.

Lines 127 and 130: I do not understand the mention of "In 15 out of the 25… in the group not tolerating a full NLT."

Line 130: A chi-squared test appears in the results section but is not mentioned in the methods.

Line 132: T30 appears twice (typo T20?)

Figure 2 is not informative.

Tables 6 and 7 are too large to be read on the page.

Table 7: PH (typo OH?)

Lines 166 to 168: Are the symptoms during the NLT or in daily life since the onset of long Covid?

Line 192: Remove the extra space

Author Response

Reviewer 3 comments:

"The authors conducted a retrospective study on a cohort of 100 patients with long Covid. The study involved administering an orthostatic tolerance test, specifically the Nasa lean test, and a C19-YRS questionnaire to the patients. The aim was to investigate the potential link between the Nasa lean test results and reported orthostatic intolerance symptoms on the questionnaire. The authors observed a significant number of patients with an abnormal Nasa lean test, which was not predictable based on the questionnaire results. Based on their findings, the authors recommend that orthostatic tolerance should be systematically evaluated in patients with long Covid. The study is noteworthy due to the inclusion of a large patient population, interesting data, and the demonstration that bedside assessment of orthostatic tolerance can be easily performed in this population.

The study, however, could be improved based on major and minor comments.

Major

One major limitation of the study is Berkson's bias, as the population that sought medical evaluation or advice at the long Covid service may not be representative of the overall population of long Covid patients. The authors should acknowledge this point and provide a more precise description of the population. For example, they should indicate whether the 100 patients were consecutive patients or how they were selected, how Covid was confirmed, how they defined severe ongoing long Covid symptoms (line 66), and what the delay was between the Covid diagnosis and the visit to the long-Covid service."

Response: Many thanks for your helpful and insightful comments. We have endeavoured to carry out the changes as proposed and have detailed the additions below.

We have added in further detail about the population studied, including how the patients were referred to the long covid service, and the method of selection of the patients (these were consecutive patients). We have expanded on the limitations discussion section, which includes a lengthy time between diagnosis of covid and the NASA Lean test.

"The authors used a simple test to assess orthostatic tolerance, which can be performed in any medical center. This test seems appropriate for evaluating the large number of patients with long Covid and contrasts with the highly specialized tests typically used to assess orthostatic tolerance. The authors should discuss this point."

Response: The introduction section now includes further details of more specialised tests such as head up tilt test and doppler ultrasound to measure cerebral blood flow, which may be carried out in more specialised centres. We have highlighted that these may not be easily accessible in many low- and middle-income countries.

"It is important to clarify for readers that there is a difference between suspected dysautonomia in patients with long Covid, based on symptoms related to the autonomic nervous system, and a demonstrated dysautonomia diagnosed using tools such as the Ewing's score."

Response: We have included a statement about dysautonomia in LC, contrasting it from conditions with known structural problems leading to dysautonomia.

"The results section includes a description of findings according to gender, but this is not described in the methods section and does not match the study objectives. The authors should address this inconsistency."

Response: The methods section now includes a phrase justifying the analysis by gender.

"Lastly, the term "cohort" is used twice in the manuscript (L 175 & L230), but it may not be appropriate as it refers to a specific biomedical research design. It is recommended to avoid using this term."

Response: We have removed the misleading word “cohort” from the manuscript.

"Minor

The authors should clearly mention early in the manuscript and in the abstract that the study is retrospective.

In lines 17 and 18, the authors mention statistical analyses, but the reader does not know what was compared. A correlation is not a comparison. Mann-Whitney's test is for unpaired data while the same patient performed NLT and C19-YRS. More generally, the statistics raise some questions: Mann-Whitney's test is for numerical data, not for categories. Pearson's correlation is parametric while Mann-Whitney's test is nonparametric."

Response: We have stated in the abstract that the study uses retrospective data. The specifics for the analysis have also been clarified in the abstract. Given the survey data for symptom severity is non normally distributed, we have re analysed the data using Spearman’s rank.

"Line 20: What does PH mean? (OH?)

Palpitations could be seen in cases of PoTS, but they could also indicate a serious cardiac disease. How were the patients with palpitations found on the C19 YRS questionnaire managed?

Table 3 mentions a maximum sustained HR change of -5 BPM. This finding is surprising in this population and could indicate a mistake or a severe dysautonomia. How were the patients with this finding managed?"

Response: We have stated in the abstract that the study uses retrospective data. The analysis has also been clarified in the abstract. Given the survey data for symptom severity is non normally distributed, we have re analysed the data using Spearman’s rank.

Patients with serve palpitations are discussed in a weekly multidisciplinary meeting which includes a consultant cardiologist, and consideration is made to investigate such patients with cardiac monitoring. We agree that a postural decrease in heart rate would be unexpected, and indeed the patient in question did have OH. We would manage by reviewing symptoms, medications, providing lifestyle advice including salt intake, and consider medical interventions if necessary.

"Lines 123 and 124: Please specify whether the OH was systolic, diastolic, or both."

Response: We have added to this section whether this was for systolic BP, diastolic BP, or both.

"Lines 127 and 130: I do not understand the mention of "In 15 out of the 25… in the group not tolerating a full NLT."

Line 130: A chi-squared test appears in the results section but is not mentioned in the methods.

Line 132: T30 appears twice (typo T20?)"

Response: The sentence previously on line 132 contained a typo and now reads “1 for PoTS at the T30 threshold, 2 at T25 and 3 at T20. “ The statement "In 15 out of the 25… in the group not tolerating a full NLT." has been clarified to “15 out of the 25 had symptoms suggestive of OI…” We have removed the statement about the chi squared test.

"Figure 2 is not informative.

Tables 6 and 7 are too large to be read on the page.

Table 7: PH (typo OH?)

Lines 166 to 168: Are the symptoms during the NLT or in daily life since the onset of long Covid?

Line 192: Remove the extra space"

Response: Figure 2 shows the distribution of the tolerance of NLT, with most patients most patients tolerating 7 minutes of lean testing. We feel this is relevant to demonstrate, given that shortened lean test may lead to under diagnosis of OI.

We have clarified that line 166 refers to symptoms experienced during the NLT.

We have carried out the formatting changes as suggested. We look forward to your reply.

Reviewer 4 Report (New Reviewer)

This is an interesting report by Isaac et al.

However, some concerns are listed below.

Were findings in adolescents different in any way than those in adults/ elderly patients?

Maybe it would be useful to ad in the discussions section a few words about the incidence of POTS and OH in the general population before the Covid pandemic.

I believe the authors should try to obtain the data regarding the symptoms recorded during NLT for the 50 participants in whom this data is missing or exclude these patients from the analysis alltogether.

Author Response

Reviewer 4 comments:

"This is an interesting report by Isaac et al.

However, some concerns are listed below.

Were findings in adolescents different in any way than those in adults/ elderly patients?"

Response: Thank you for your helpful comments. Our study aim was not to compare different groups of patients with long covid, but rather to get a feel of overall trends in OI in LC patients in a community LC service. Our study numbers were too small to compare between different age categories, with only one patient under 20, and no patients over the age of 74. We therefore were unable to compare results by age.

"Maybe it would be useful to ad in the discussions section a few words about the incidence of POTS and OH in the general population before the Covid pandemic."

Response: Thank you for your comment regarding OI in the general population. We have added in some prevalence figures in the general population. It is clear that rates of OI in the general population vary significantly by age and gender, and it is one of the limitations of our study that we weren’t able to perform subgroup analysis by age due to the small sample size.

"I believe the authors should try to obtain the data regarding the symptoms recorded during NLT for the 50 participants in whom this data is missing or exclude these patients from the analysis alltogether."

Response: Data on the 50 patients is not missing. There were no recorded symptoms. The normal process is for patients symptoms to be recorded during the NLT, and we feel it is a reasonable assumption that these patients were asymptomatic during the NLT.

Round 2

Reviewer 1 Report (Previous Reviewer 1)

I would still recommend the article to take a different turn. as nasa lean test can detect in a probably less then halve of the orthostatic intolerance population hemodynamic differences, taking OI complaints present more serious then hemodynamic HR and BP changes can suggest and therefore basing conclusions on presence or absence of OI NOT on nasa lean test results but a comination of results and complaints. Passive standing in HUT will always have a greater potential in showing positive OI results. So i would suggest rewriting this advice in introduction, discussion and especially conclusion is needed. This to clarify for the readers that have no experience in this field. Please sent for review of the update.

Author Response

Thank you again for your comments. We agree that HUT will have a greater potential in showing positive OI results.

We have already stated in the Introduction section “In contrast, other more specialised tests for diagnosing OI in the context of LC include Head Up Tilt (HUT) testing and Cerebral Blood flow (CBF) through transcranial doppler. As it provides a more direct measurement of cerebral perfusion, which is assumed to lead to symptoms of OI, CBF may be a more sensitive test for OI.”

We have clearly stated the limitations of these specialised tests: “access to HUT or CBF is limited to large hospital and research settings and is not feasible to be used in all patients in a clinic setting and in Low- and Middle-Income Countries (LMIC).”

If we do not have simple clinic-based tests for detecting OI, there is a huge risk that OI in a large number of individuals will be missed. Lean test is therefore recommended for using in a clinic setting in Long Covid by NICE and NHS England.

We have also reviewed available tests other than the Lean Test in the Discussion section, particularly Cerebral Blood Flow (CBF) testing, which has shown to be useful in the context of patients with normal HUT with OI symptoms. We have stated that further work is required to evaluate the sensitivity and specificity of Lean test in the Long Covid by subjecting a large number of patients to CBF in a research study.

We hope this clarifies that we are in agreement with you and have already added all the points you have raised in the relevant sections of the manuscript.

Reviewer 4 Report (New Reviewer)

I feel the authors have adequately adressed my comments.

Author Response

We appreciate your time and feedback.

This manuscript is a resubmission of an earlier submission. The following is a list of the peer review reports and author responses from that submission.

Round 1

Reviewer 1 Report

The definitions described for pots are different than the ones originating from the 2015 heart rhythm society; those are considered standard. The age for >40 is up to 19, 18 or less, not 20. Maybe using 2011 data where more recent definitions are available is not suitable.

Looking at T20 and T25 as pots alternatives is not according to definitions and should be referred to as a normal heart rate and blood pressure response. In healthy controls of a reference not used for 18 bpm during end-tilt, pushing patients with orthostatic complaints into a pots diagnosis is inappropriate. WIth the large number of patients with hemodynamic abnormalities but still having a large amount of complaints using this test instead of a test with transcranial doppler or extracranial doppler of which techniques a lot of information is available, a big proportion of patients will be misdiagnosed as having not so much wrong.

The test might be suitable for GP or first screening, but in specialized OI clinics a more extensive test also checking cerebral blood flow (velocity) is scientifically more sound. Thereby not dismissing patients with OI complaints but without hemodynamic abnormalities on leening test or standing test.

Author Response

Thank you for taking the time to read through our work and for your comments. We have updated our reference to take into account the 2015 heart rhythm society PoTS definition.

We have retained the PoTS standard agreed criteria as per 2015 PoTS consensus statement. We have clarified that the additional analysis on patients with T25 and T20 is for values at the upper limits of normal values for dysautonomia, rather than a PoTS definition.

We currently do not offer cerebral blood flow imaging to patients in this long covid service. Thank you once again and we look forward to your response.

Reviewer 2 Report

Thank you for the opportunity to review this interesting manuscript which reports on orthostatic intolerance in patients with long covid. Using simple methodology, the authors report some interesting finings that help improve understanding of the effect of COVID on cardiovascular system.

There are several comments that need to be addressed in order for the manuscript to be improved.

1. Change title to 'Orthostatic intolerance in patients living with long covid'.

2. Methodology: it is not clear was this prospective or retrospective study.

3. Methodology: describe details of the NASA lean test.

4. Results: add tables with mean +/- SD (and range) for haemodynamic measures heart rate and blood pressures

5. Results: Gender is an important confounding factor. Analyse and present data (mean +/- SD) for gender differences for main measures ie HR, BPs, C19-YRS scores. Compare i) men vs women, and ii) prevalence of main outcomes according to gender despite groups (N) is not equal between men and women.

6. Results: Present significant relationships in the figure using scatter dots graph. 

7. Discussion and Conclusion. Update to reflect additional analyses and gender differences.

Author Response

Thank you for taking the time to review our work, and for your helpful comments. We have responded to the suggested changes as below.

1. Change title to 'Orthostatic intolerance in patients living with long covid'.

Response: Thank you for the suggested title, which takes into consideration our usage of C19-YRS data. We have changed this slightly to remove the word “patient”, to “Orthostatic intolerance in long covid”.

2. Methodology: it is not clear was this prospective or retrospective study.

Response: The study is retrospective, and line 70 in the materials and methods section states that the data was extracted retrospectively. We have also clarified this in the abstract.

3. Methodology: describe details of the NASA lean test.

Response: Further details of the NLT have been added to the methodology section.

4. Results: add tables with mean +/- SD (and range) for haemodynamic measures heart rate and blood pressures

Response: Thank you. We have now added Table 3 to show  mean +/- SD values.

5. Results: Gender is an important confounding factor. Analyse and present data (mean +/- SD) for gender differences for main measures ie HR, BPs, C19-YRS scores. Compare i) men vs women, and ii) prevalence of main outcomes according to gender despite groups (N) is not equal between men and women.

Response: We have presented the data by gender, to include numbers fitting the hemodynamic criteria for PoTS by gender, as well as the C19-YRS data (see Tables 4 and 5 in the revised manuscript)

6. Results: Present significant relationships in the figure using scatter dots graph. 

Response: We have found no significant relationships in our results.

7. Discussion and Conclusion. Update to reflect additional analyses and gender differences.

Response: We have updated the discussion to include that there were no significant differences in analysis by gender.

Thanks again for your comments, and we look forward to your reply.

Round 2

Reviewer 1 Report

My comments on the first review are not properly addressed. Nasa lean test does NOT check for cerebral blood flow and flow velocity during the upright position and therefore only can be a screening test for the general practitioner. It should not be the test to go for in an orthostatic intolerance specialized clinic.

Author Response

We agree with reviewer’s comments and thank them for their observations. This study is related to a screening test (NASA lean test) for use in the community by general physicians. The study did not take place in a specialist centre having the capacity to carry out confirmatory tests such as cerebral blood flow. Given the prevalence of Long Covid (>50 million worldwide), it is impractical for every individual to have a cerebral blood flow test or other confirmatory tests. The National Institute for Clinical Excellence recommends using screening test such as active stand or lean test for evaluation of dysautonomia in all Long Covid clinics. There is an urgent need for evaluation of the screening test that can be done on all individuals with Long Covid and this has been explored for the first time in the literature in this study. We agree that the patients who screen positive for OI must then be referred to specialist dysautonomia services for further specialist confirmatory tests such as Tilt Table test or cerebral blood flow test. We have therefore revised the entire manuscript to clarify that NASA lean test is a screening test (and not a confirmatory test) for OI. We have also modified the conclusion to clarify that further work is needed to detect the sensitivity and specificity of the screening test.